# On the Learning Mechanisms in Physical Reasoning

**Shiqian Li**[⋆,1,2,5], **Kewen Wu**[⋆,3,5], **Chi Zhang**[4,5 ✉], **Yixin Zhu**[1,2 ✉]

[1] School of Intelligence Science and Technology, Peking University
[2] Institute for Artificial Intelligence, Peking University
[3] Department of Automation, Tsinghua University
[4] Department of Computer Science, University of California, Los Angeles
[5] Beijing Institute for General Artificial Intelligence (BIGAI)

Project Website https://lishiqianhugh.github.io/LfID_Page

## Abstract

Is dynamics prediction indispensable for physical reasoning? If so, what kind of roles do the dynamics prediction modules play during the physical reasoning process? Most studies focus on designing dynamics prediction networks and treating physical reasoning as a downstream task without investigating the questions above, taking for granted that the designed dynamics prediction would undoubtedly help the reasoning process. In this work, we take a closer look at this assumption, exploring this fundamental hypothesis by comparing two learning mechanisms: Learning from Dynamics (LfD) and Learning from Intuition (LfI). In the **first experiment**, we directly examine and compare these two mechanisms. Results show a surprising finding: Simple LfI is better than or on par with state-of-the-art LfD. This observation leads to the **second experiment** with Ground-truth Dynamics (GD), the ideal case of LfD wherein dynamics are obtained directly from a simulator. Results show that dynamics, if directly given instead of approximated, would achieve much higher performance than LfI alone on physical reasoning; this essentially serves as the performance upper bound. Yet practically, LfD mechanism can only predict Approximate Dynamics (AD) using dynamics learning modules that mimic the physical laws, making the following downstream physical reasoning modules degenerate into the LfI paradigm; see the **third experiment**. We note that this issue is hard to mitigate, as dynamics prediction errors inevitably accumulate in the long horizon. Finally, in the **fourth experiment**, we note that LfI, the extremely simpler strategy when done right, is more effective in learning to solve physical reasoning problems. Taken together, the results on the challenging benchmark of PHYRE [3] show that LfI is, if not better, as good as LfD with bells and whistles for dynamics prediction. However, the potential improvement from LfD, though challenging, remains lucrative.

## 1 Introduction

Humans possess a distinctive ability of understanding physical concepts and performing complex physical reasoning. The literature on humans learning mechanisms for solving physical reasoning problems can be categorized into two schools of thought [43]: (i) physical intuition at a glance without much thinking, such as judging whether a stacked block tower will collapse [6], and (ii) more

---

⋆ indicates equal contribution.
✉ indicates corresponding authors.

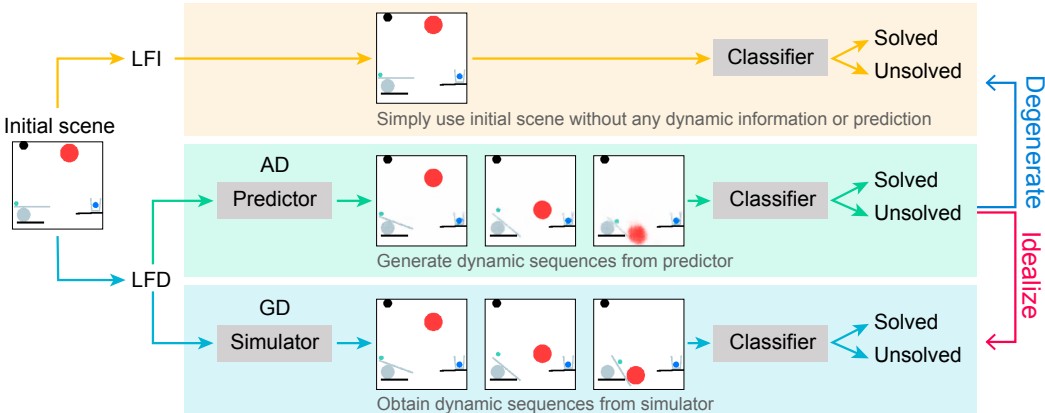

Figure 1: **Comparison of the two learning mechanisms.** Learning from Intuition (LfI) learns from intuition by directly using a classifier (first row in yellow). Based on the source of dynamics, we divide Learning from Dynamics (LfD) into learning with Approximate Dynamics (AD) (second row in green) and Ground-truth Dynamics (GD) (third row in blue). Specifically, learning with GD leverages ground-truth dynamics from the simulator, whereas learning with AD predicts how the objects' positions and poses unfold via a dynamics predictor. In theory, learning with AD should reach the same performance as learning with GD if dynamics prediction were perfect; *i.e.*, learning with GD can be regarded as the ideal case of learning with AD. However, in practice, learning with AD usually degenerates into LfI due to very inaccurate prediction.

extensive unfolding of states under the assumed physical dynamics when facing complex physical tasks [7]. Such a disparity between System-1-and-System-2-like problem-solving strategies [36] motivates us to think over the learning mechanisms for physical reasoning in machines:

> *When performing physical reasoning, is it better for machines to learn from intuition by simply analyzing the static physical structure, or to learn from dynamics by predicting future states?*

Recent physical reasoning benchmarks for machine learning are mostly physics engines focusing on evaluating the task-solving abilities of models. For example, PHYRE [3] and Virtual Tools [1] contain physical scenes with long-term dynamics and complex physics interactions. These environments have various tasks with explicit goals, such as making the green object touch the blue object by *placing a new red ball* into the initial scene. An artificial agent is tasked to predict the final outcome, *e.g.*, whether the placed *red ball* will successfully solve the given task.

Existing problem-solving methods approach such physical reasoning problems by designing various future prediction modules [23, 53]. These modules are devised under the assumption that human brains inherently possess a simplified physics engine (called intuitive physics engine) [18], akin to a computer game simulator, capable of predicting objects' future states and changes.

Although the intuitive theory claims that humans can predict physical outcomes rapidly, it does not directly guide at the computational level more than the hypothesis that we might have a physics-engine-like mechanism in our brain [18, 43]. Critically, although humans can predict dynamics under the intuitive theory, dynamics prediction might not be necessary for all types of physical reasoning tasks. This hypothesis is largely left untouched, especially at the computational level. Just as noted by Lerer et al. [46], directly learning by intuition without dynamics prediction is sufficient for various physical reasoning tasks. Of note, this hypothesis does not contradict the intuitive theory but rather provides a new perspective at the computational level.

In this paper, we conduct a series of experiments to answer the above questions empirically. To the best of our knowledge, ours is the first to **systematically** compare the LfI and LfD paradigms. In the **first experiment**, we verify the simple approach of LfI by training a classifier [15] to predict whether an action would lead to success in problem-solving. Surprisingly, in the preliminary study, such a model already reaches the state-of-the-art (SOTA) performance and even outperforms existing LfD methods in unseen scenarios, indicating better generalization. Inspired by this counter-intuitive result, we conduct more experiments on the two learning mechanisms; see Fig. 1 for an illustration. In the **second experiment**, we first set out to investigate whether LfD could work better than LfI in theory by measuring the performance of a video-based classifier [8] using the ground-truth dynamics,

setting an upper bound for the route of dynamics prediction. In the **third experiment**, we replace the ground-truth dynamics with predicted dynamics from an advanced future prediction model [62] to see how LfD performs in practice. The results from these two experiments reveal that precise dynamics significantly improve model performance, but the predicted approximations fail to work as expected: Approximation brings disturbing biases and causes the performance to degenerate into LfI or even worse in certain tasks. In the **fourth experiment**, through a series of experiments on various LfI models, we conclude that LfI could be a simpler and more practical paradigm in physical reasoning. However, making breakthroughs in physical prediction is still promising though challenging. We hope that our discussions shed light on future studies on physical reasoning.

## 2 Related Work

**Intuitive physics and physical reasoning** Since Battaglia et al. [6], the computational aspects of intuitive physics have attracted research attention [43, 75]; intuitive physics and stability has since been further incorporated in complex object [77, 64, 49, 17, 50, 72] and scene [74, 65, 73, 76, 47, 30, 9, 63], and task [26, 27, 32] understanding tasks. The progress enables machines to learn to judge (i) which object is heavier after observing two objects collide [21, 54, 59], (ii) whether a stacked block tower will fall [6, 24, 46], (iii) whether water in two different containers will pour at the same angle if tilted [41, 57] or liquid in general [5], and (iv) behaviors of dynamics with various materials [42]. However, this line of work primarily focuses on physical tasks without long-term dynamics, either by knowledge-based approaches [6, 54, 70] or learning-based approaches [24, 46].

More complex physical reasoning problems [1, 3, 69], including those involving question answering [10, 11, 14, 29, 71], have also been studied. In particular, Allen et al. [1] propose to use knowledge-based simulation; Xu et al. [68] adopt a Bayesian symbolic method; Battaglia et al. [7], Girdhar et al. [23], and Qi et al. [53] recruit the graph-based interaction network. However, none of these methods fully justify the necessity of dynamics prediction. In this work, we challenge this fundamental assumption and point out a simpler and efficacious but overlooked solution.

**PHYRE and relevant environments** Bakhtin et al. [3] introduce the novel physical reasoning task of PHYRE, wherein an agent is tasked with finding action in an initial scene to reach the goal under physical laws. Current methods for solving PHYRE include reinforcement learning (*e.g.*, DQN [3]), forward prediction neural networks with pixel-based representation [23], and object-based representation [23, 53]. Notably, Girdhar et al. [23] adopt different kinds of forward prediction architectures to perform PHYRE tasks but fail to obtain significant performance improvement, whereas Qi et al. [53] design convolutional interaction network to learn long-term dynamics, achieving SOTA performance by leveraging ground-truth information about object states in the physical scenes.

In fact, the physical reasoning task of PHYRE can be regarded as an image classification task on judging whether an initial action would lead to a successful outcome, or a video classification task by considering the dynamics after the initial action is performed. The success of the former sat on deep convolutional neural networks [28, 40] and has now shifted to Transformer-based models [4, 15, 51]. The change from convolutional architectures to attentional models also inspires recent advances in video classification. Models such as TimeSformer [8] and ViViT [2] in this domain also expand into fields such as action recognition [22] and group activity recognition [20].

Unlike PHYRE which directly focuses on physical reasoning in a simplified virtual environment, some benchmarks include physical reasoning in their environments more implicitly and take physics as an aid to finish tasks in real life, such as the ones in autonomous driving [16] and embodied AI [39, 67, 19, 55, 48]. Robotic controller based on physics engines [25, 45, 38, 12, 60], navigation tasks on 3D physical scenes [66], and more broadly task and motion planner [61, 37, 33, 35, 34] may also need physical understanding modules in the system.

**Dynamics prediction** Predicting dynamics into the future is one of the most extensively studied topics in the vision community. One modern approach is to extract image representation and incorporate an RNN predictor [58, 62] or a cycle GAN-based approach [44]. However, these approaches cannot extract robust representation from the pixels and incur accumulated errors in long-term prediction. To tackle this problem, Janner et al. [31] and Qi et al. [53] focus on object-centric representation; these are task-specific solutions with various inductive biases (*e.g.*, spatial information, the number of objects), and the performance drops when dealing with multiple objects with occlusions [52].

## 3   The Two Learning Mechanisms

In this section, we define the two learning mechanisms for solving physical reasoning problems. Henceforth, we denote all objects' states at time $t$ as $X_t$. Given an initial background image $I$ of a physical setup and a random distribution of actions $\mathcal{A}$, the model needs to learn a distribution of the final outcome $P(y|X_0)$, where $X_0 = \{\mathcal{A}, I\}$, and $y$ denotes the possible outcome.

**Mechanism 1. Learning from Intuition (LfI)**   In LfI, the outcome $y$ is learned directly from the initial images and actions using a task-solution model $f$:

$$P(y|X_0) = f(X_0; \theta), \tag{1}$$

where $\theta$ denotes the parameters of the task-solution model $f(\cdot)$. We call this mechanism LfI because $f(\cdot)$ can be viewed as an intuitive map from the initial conditions to the outcome.

**Mechanism 2. Learning from Dynamics (LfD)**   The nature of physics is inherently dynamic. As such, in LfD, $y$ is no longer directly learned from the initial scenes; instead, this approach first learns the underlying dynamics $D = \{X_t | t = 0, 1, \ldots, T\}$ within a time window $T$ using a dynamics prediction module $g(\cdot)$, and then predicts the outcome from the predicted dynamics. Formally, the forward process is described as below:

$$P(y|X_0) = f(D; \theta), \text{ where } D = g(X_0; \phi), \tag{2}$$

where $\phi$ represents the parameters of the dynamics prediction model $g(\cdot)$. Usually, $g(\cdot)$ is implemented as either an auto-regressive module based on pixel presentation or a graph-based interaction network from object-based representation. In this work, we consider two optimization schedules for $g(\cdot)$: parallel optimization during joint learning of $f(\cdot)$ or serial optimization by learning and fixing $g(\cdot)$ first; please refer to Algs. 1 and 2 for more details.

---

**Algorithm 1:** Parallel optimization of LfD

**Variables:**
  $I$ is the initial background image. $\mathcal{A}$ is the action. $f(\cdot)$ and $g(\cdot)$ are the task-solution model and the dynamics prediction model, respectively. $D$ and $y$ denote predicted dynamics and outcome, and $D_{gt}$ and $y_{gt}$ the ground-truth ones. The dynamics loss and cross-entropy loss are denoted as $L_d(D, D_{gt})$ and $L_e(y, y_{gt})$, respectively. $\alpha$ and $\beta$ are hyperparameters to balance the two losses.
1: **repeat**
2:    Predict the dynamics $D$ from $I$ and $\mathcal{A}$ using $g(\cdot)$;
3:    Predict the outcome $y$ from $D$ using $f(\cdot)$;
4:    Compute the total loss $L_{total}(D, y, D_{gt}, y_{gt}) = \alpha L_d(D, D_{gt}) + \beta L_e(y, y_{gt})$;
5:    Optimize $f(\cdot)$ and $g(\cdot)$ simultaneously using the gradient of the total loss $\nabla L_{total}(D, y, D_{gt}, y_{gt})$.
6: **until** max iteration

---

**Algorithm 2:** Serial optimization of LfD

**Variables:**
  $I$ is the initial background image. $\mathcal{A}$ is the action. $f(\cdot)$ and $g(\cdot)$ are the task-solution model and the dynamics prediction model, respectively. $D$ and $y$ denote predicted dynamics and outcome, and $D_{gt}$ and $y_{gt}$ the ground-truth ones.
1: **repeat**
2:    Predict the dynamics $D$ from $I$ and $\mathcal{A}$ using $g(\cdot)$;
3:    Compute the dynamics loss $L_d(D, D_{gt})$;
4:    Optimize $g(\cdot)$ using the gradient of dynamics loss $\nabla L_d(D, D_{gt})$;
5: **until** max iteration
6: Freeze $g(\cdot)$;
7: **repeat**
8:    Predict the dynamics $D$ from $I$ and $\mathcal{A}$ using the pre-trained $g(\cdot)$;
9:    Predict the final outcome $y$ from $D$ using $f(\cdot)$;
10:    Compute the cross-entropy loss $L_e(y, y_{gt})$;
11:    Optimize $f(\cdot)$ using the gradient of the cross entropy loss $\nabla L_e(y, y_{gt})$;
12: **until** max iteration

---

While the dynamics prediction step and the outcome prediction step can be integrated together, it is worth noting that in LfD, additional architecture changes and supervisory signals are necessary to learn the underlying dynamics, without which the paradigm degenerates into LfI. Ideally, a physics engine or a simulator plays the role of future prediction. However, inversely learning the physical laws [56, 13] is very demanding due to the intrinsic challenges in long-term prediction.

# 4 Experimental Setup

In this section, we briefly introduce the challenging physical reasoning benchmark of PHYRE-B and the setup of our experiments. Additional training details are in the supplementary material.

**Environment** PHYRE-B is a goal-driven benchmark consisting of 25 different task templates of physical puzzles that can be solved by placing a red ball (hence the "B"). Each template has 100 similar tasks, which enables two different evaluation settings. (i) Within-template setting: train on 80% of tasks of each template and test on the rest 20% of each template. (ii) Cross-template setting: train on all tasks of 20 from the 25 templates and test on the rest of five previously unseen templates.

The performance is evaluated by AUCCESS [3], a weighted sum of the accuracy rate related to the number of attempts. To encourage as few attempts as possible to solve a puzzle, the weights are calculated as: $\omega_k = \log(k+1) - \log(k)$, where $k \in \{1, 2, 3...100\}$ represents the range of attempts. Formally, AUCCESS $= \frac{\sum_k \omega_k \cdot s_k}{\sum_k \omega_k}$, where $s_k$ is the success rate with $k$ attempts.

This physical reasoning benchmark is deemed challenging primarily due to the following three reasons: (i) The environment involves a variety of objects of different shapes and mass distribution, such as balls, bars, standing sticks, and jars. (ii) The solution set takes up only a small part of the whole action space, making randomly sampled actions hardly ever work out. (iii) Before reaching the goal, a series of complex physical interactions would happen, involving falling, rotation, collision, and friction. To the best of our knowledge, no success has been claimed on the environment.

**Experiment 1: LfD *vs*. LfI** In the first experiment, we compare the SOTA LfD model on PHYRE and a transformer-based classifier for LfI. Specifically, we pick the RPIN model [53] to represent LfD and the ViT model [15] for LfI. The RPIN model leverages a convolutional interaction network based on object-centric representation and predicts the object states (bounding boxes and masks) into the future. When solving a PHYRE task, the model first predicts 10-time steps into the future for an action and then recruits an MLP-based task-solution model to predict the outcome. In comparison, ViT is a transformer-based classifier initially designed for image classification. The model patches an image, decomposes it into non-overlapping regions, and performs a series of self-attention. The image-level representation is finally extracted by appending a CLS token.

**Experiment 2: LfD under GD** The second experiment serves as a diagnostic test for the efficacy of dynamics in physical reasoning tasks. To this end, we directly extract simulator results and feed the image sequence into a video classifier model, setting an upper bound for the LfD paradigm by replacing dynamics prediction with the ground truth. In this experiment, we adopt the TimeSformer model [8] due to its superior performance. The model patches the image sequence and replaces traditional convolution architectures with interleaving spatial attention and temporal attention.

**Experiment 3: LfD under AD** We dive deep into the ineffectiveness of LfD methods like RPIN by fixing the task-solution model with TimeSformer but replacing the input with trajectories predicted by a learned module. In particular, we train a PredRNN [62] to learn the dynamics. PredRNN is an advanced pixel-based video predictor extending the inner-layer transition function of memory states in LSTMs to zigzag memory flow that propagates in both bottom-up and top-down directions. A curriculum is set up during learning for better long-term prediction [62].

**Experiment 4: More on LfI** In the final set of experiments, we verify the performance of different LfI methods. In addition to ViT, we consider BEiT [4] and Swin Transformer [51], which have been proven beneficial in image classification tasks.

# 5 Comparison of Learning from Intuition and Learning from Dynamics

In this section, we start our discussion with a preliminary experiment and consequently explore whether dynamics contribute to better judgments in physical reasoning tasks both in ideal conditions and in reality by comparing the performance of LfI and LfD.

## 5.1 LfD *vs*. LfI: A surprising finding

To tackle the demanding physical puzzles in PHYRE-B, previous studies use pixel-based or object-based future prediction modules to boost problem-solving performance [23, 53]. The SOTA LfD

method RPIN uses a convolutional interaction network to predict long-term dynamics. Inspired by previous studies on intuitive physics, we set out to see whether a decision can be made directly by a classifier of ViT. Counter-intuitively, while the supervision from the dynamics should ideally help a model in LfD look into the future, the performance of the LfI model reaches better than or on par with SOTA as shown in Tab. 1. The AUCCESS of ViT in the cross-template setting even significantly outperforms RPIN's, demonstrating its high generalization ability when solving unseen physical puzzles. Driven by this surprising and intriguing finding, we conduct additional follow-up experiments to determine if the dynamics prediction component helps physical reasoning.

Table 1: **The performance of RPIN and ViT in solving PHYRE-B puzzles.** We report the AUCCESS in both within and cross settings. RPIN uses objects' information as prior knowledge and extra supervisory signals for dynamics prediction, whereas ViT is trained only supervised by the final outcomes with initial scenes as input.

| Model | Mechanism | Input | Supervision | Within | Cross |
|-------|-----------|-------|-------------|--------|-------|
| RPIN | LfD | Initial scenes, bboxs | Bboxs, masks, outcomes | 85.49 | 50.86 |
| ViT | LfI | Initial scenes | Outcomes | 84.16±0.30 | 56.31±1.95 |

## 5.2 LfD under GD: Is ground-truth dynamics better than intuition?

The surprising discovery motivates us to ask why the LfD model has similar or even worse performance than LfI. As a first step, we question the merit of dynamics: Do dynamics help solve these physical puzzles? To answer this question, we supply the model with the best-case—ground-truth—dynamics and provide the information to our task-solution model of TimeSformer, due to the video-classification-like nature. We vary the number of time frames supplied to the model: We consider the input of lengths 1, 2, 4, and 8 extracted directly from the simulator with a time interval of 1 second. It is worth noting that using GD, we assume an ideal dynamics prediction model $g(\cdot)$ that accurately predicts the future.

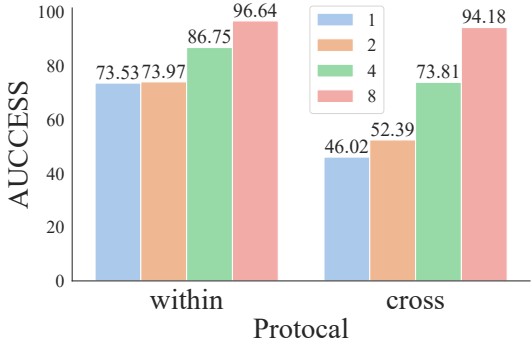

Figure 2: **Performance of TimeSformer on PHYRE with different ground-truth input lengths.** We compare the AUCCESS of within-template and cross-template evaluation settings.

The performance of LfD with GD is shown in Fig. 2. While the model underperforms ViT with fewer than 3 frames, which is out of the scope of our work, we see that AUCCESS is significantly boosted with four or more frames. Specifically, with 8 input frames, AUCCESS can be improved by 23.11 in the within setting and 48.16 in the cross setting, reaching about 95% in both settings. This observation answers the above question: Dynamics *do* help problem-solve in physical reasoning. Such an observation aligns with our intuition that dynamics can tell more about the future states, thus making learning decisions easier for the task-solution model. With sufficient dynamics information, the problem could be "solved" to a large extent.

However, if dynamics *do* help, why do current LfD methods fail to outperform LfI as shown in Sec. 5.1, even considerably lower than the simple baseline in generalization? Is it because of inaccurate dynamics prediction or because of the outdated design in task-solution models? We answer this question in the following set of experiments.

## 5.3 LfD under AD: How do approximate dynamics perform?

Given the fact that GD does help problem-solving, we explore the reason why AD—the dynamics prediction in practice—does not take effect as expected. Specifically, we fix the task-solution model $f(\cdot)$ with TimeSformer [8] as in Sec. 5.2 and instantiate the dynamics prediction model $g(\cdot)$ with an advanced video prediction model PredRNN [62]. We train the LfD pipeline using both of the two optimization schedules mentioned in Sec. 3 to investigate which schedule can better help in learning useful dynamics information.

The PredRNN first takes one initial frame as the input and outputs three predicted future frames. Next, the TimeSformer takes all the frames as input, including the first initial frame and the three predicted frames, and outputs the final outcome. For parallel optimization, we train an end-to-end framework by integrating the output from PredRNN with the input of TimeSformer. The model parameters $\phi$ and $\theta$ are updated simultaneously by backpropagating both the dynamics-learning loss and the final cross-entropy loss. For serial optimization, the PredRNN is trained first independently using ground-truth future dynamics as supervision and kept fixed, after which the TimeSformer is optimized using the output from the pre-trained PredRNN, supervised by the ground-truth decisions. For both optimization schedules, the PredRNN only takes raw RGB images as the input without extra information, such as bounding boxes or masks of each object, and directly generates the predicted future frames without further rendering.

We report the AUCCESS of both optimization schedules on PHYRE-B in Tab. 2. These results show that independent of the optimization schedule used, AD falls far behind from GD and performs equally or even worse than LfI, indicating that approximate dynamics do little help for the task-solution model in making better judgments. Specifically, by comparing the performance of parallel optimization to TimeSformer with a single frame, we observe a tendency of LfD pipeline's performance degeneration to LfI, ending up with similar AUCCESS. Moreover, Sec. 5.3 shows that for both within and cross settings, the test dynamics loss in parallel optimization is much higher than the corresponding one in serial optimization,

Table 2: **The performance of AD following two optimization schedules in LfD.** We use the same task-solution model TimeSformer for all the experiments. **NF** denotes the number of input frames used by the task-solution model. We also list the results using GD for comparison.

| Prediction | NF | Within | Cross |
|---|---|---|---|
| PredRNN (parallel) | 4 | 75.22 | 46.42 |
| PredRNN (serial) | 4 | 64.90 | 44.33 |
| / | 1 | 73.53 | 46.02 |
| Simulator | 4 | 86.75 | 73.81 |

which suggests that the representation learned by LfD pipeline in parallel optimization has difficulty in capturing accurate dynamics and is severely impacted by the cross-entropy loss, as is evident in the visualization results in Fig. 3. Although serial optimization gives the dynamics prediction model ample opportunities to learn better pixel-wise approximations of future frames, it still fails to make better decisions than parallel optimization. We hypothesize that in serial optimization, the task-solution model eventually uses the noisy predicted dynamics, which hinders the performance due to inaccuracy; in parallel optimization, the model can avoid this problem by paying less attention to learning future dynamics and focusing more on directly learning the outcome in a manner similar to LfI. Taken together, these experimental results pinpoint the reasons for the inefficacy of current LfD methods with AD: (i) It is the poor dynamics learned, rather than the task-solution model, that causes the performance degradation. Inaccuracy dynamics prevent accurate reasoning. (ii) Existing LfD methods with AD degenerate into LfI in the best case.

By comparing the models above, we conclude that dynamics prediction, in theory, does help solve physical puzzles by providing more temporal information, while struggling to show the full strengths in reality due to the challenges in handling accumulated uncertainty in a long horizon. The results in serial and parallel optimization also reveal the fact that dynamics prediction plays an important role regardless of the task-solution model, but inaccurate dy-

| Opt | Loss | Within | Cross |
|---|---|---|---|
| Parallel | entropy | 0.0638 | 0.5726 |
| | dynamics | 0.0039 | 0.0049 |
| Serial | entropy | 0.1285 | 0.6554 |
| | dynamics | 0.0003 | 0.0021 |

namics will eventually harm downstream reasoning; in the best case, LfD degenerates into LfI.

### 5.4 More on LfI: How does LfI perform?

Observing the notable success of ViT in the physical reasoning task, we consider testing additional visual classification models to verify the effectiveness of the LfI paradigm. Among a myriad of models, we pick ViT, Swin Transformer, and BEiT for experiments due to their demonstrated superiority in image classification. Of note, TimeSformer with a single input frame can also be considered as a model in the LfI paradigm. All of the models take only the first frame as the input. For equal comparison, we train all of the LfI models with an equal number of actions per task as in RPIN. Nevertheless, the LfI models take no extra prior object information as the input and use only final outcome labels as supervision signals, significantly simplifying the learning process.

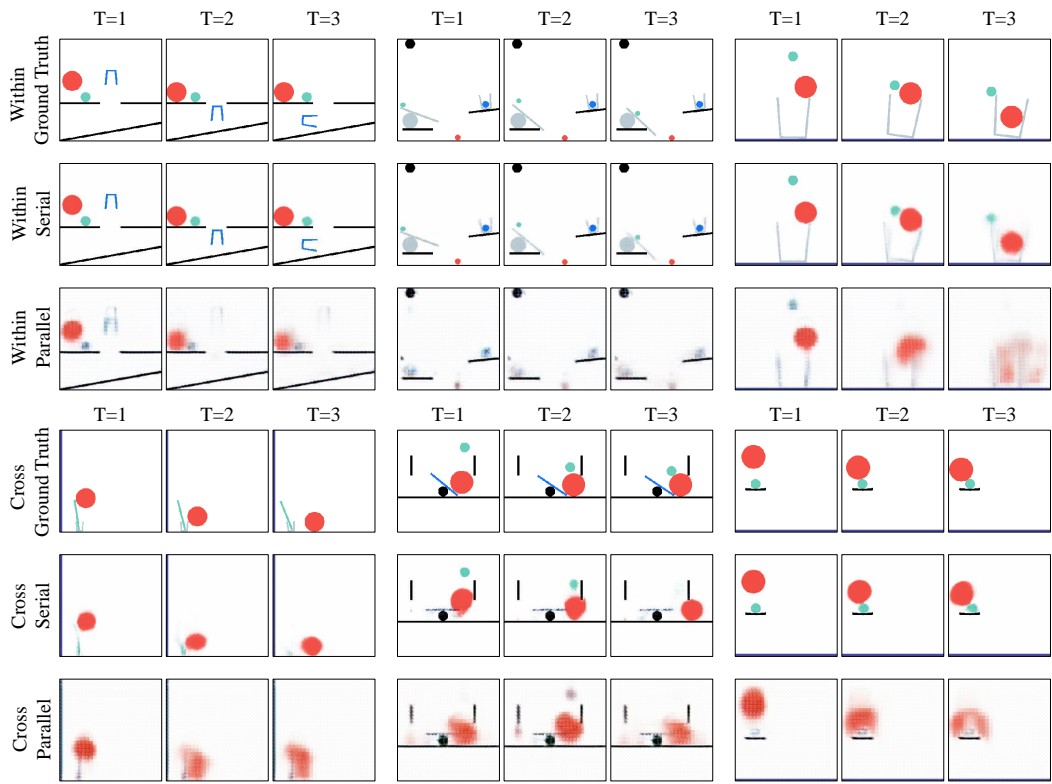

Figure 3: **Predicted dynamics from PredRNN in LfD's two optimization schedules.** In parallel optimization, PredRNN hardly learns any dynamics, whereas the dynamics learned in serial optimization are more accurate. However, serial optimization still fails to perform better than parallel optimization as shown in Tab. 2, indicating that improved but still noisy dynamics don't lead to better problem-solving. We refer the readers to the supplementary material for more results.

The results of these LfI models are presented in Tab. 3. The performance of LfI models in the within-template setting is competitive with the SOTA RPIN. In addition, all of the LfI models outperform SOTA in the cross-template setting in AUCCESS; ViT even outperforms by almost 6 points on average. By visualizing the distribution heat maps of the possible solution set learned by ViT and comparing them with the ground truth in within-template and cross-template settings (see Fig. 4), we observe that the ViT model not only achieves high performance but also accurately recovers the underlying solution distributions. We hypothesize that intuitive models, though trained without dynamics information, can still extract high-level spatial knowledge and physical commonsense that generalizes well in unseen scenarios by looking at the data only; for instance, the most suitable red ball position should be close to the moving objects to exert influence. However, we believe that the performance of LfI still has significant room for improvement since the inductive biases and structure designs for physical reasoning are not explicitly considered.

Table 3: **Performance of different LfI models in solving PHYRE-B problems.** For ease of comparison, we also list results from the previous SOTA. We report the AUCCESS in within-template and cross-template settings. Input to ViT, Swin Transformer, and BEiT are only the first frame. For Dec [Joint] and RPIN, we directly use their reported AUCCESS for comparison.

| Model | Mechanism | Object Info | Supervision | Within | Cross |
|---|---|---|---|---|---|
| ViT | LfI | False | Outcome | 84.16±0.30 | **56.31±1.95** |
| Swin | LfI | False | Outcome | 84.71±0.33 | 54.92±2.30 |
| BEiT | LfI | False | Outcome | 83.59±0.09 | 54.07±1.88 |
| Dec [Joint] | LfD under AD | False | Dynamics & Outcome | 79.73 | 52.64 |
| RPIN | LfD under AD | True | Dynamics & Outcome | **85.49** | 50.86 |

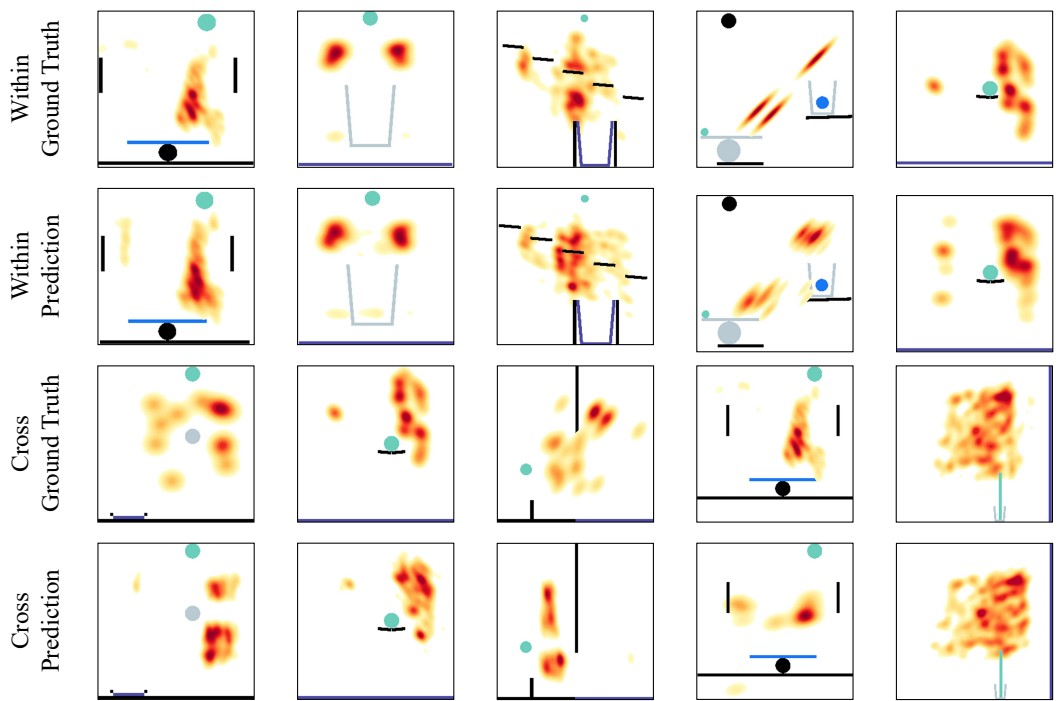

Figure 4: **The ground-truth $P(y|X_0)$ distribution heat maps and the ones predicted by ViT in PHYRE-B.** The maps are generated using the first 10,000 actions from the simulation cache offered by PHYRE-B. We only map the 2D positions of the placed red balls without showing their radii. For prediction heat maps, only actions with a likelihood above 0.8 are used for visualization. A warmer area denotes a higher likelihood of success. ViT learns an accurate solution set as the ground truth. We refer the readers to the supplementary material for the maps of Swin and BEiT.

Besides the promising performance, LfI models also demonstrate the following advantages:

- They are design-efficient without complex hand-crafted tweaks for dynamics prediction modules, which in some cases only introduce harmful distraction or noisy representation; see Sec. 5.3.

- They only take initial scenes as the input and require no extra task-specific prior knowledge about the objects as used in object-centric dynamics prediction.

- They can be easily pre-trained on other computer vision tasks (*e.g.*, image classification on ImageNet), incorporating general domain-agnostic representation to avoid task-specific overfitting.

In summary, we view LfI as a simpler and more effective paradigm for physical reasoning.

## 6  Conclusion and Discussion

We introduce the concepts of two learning mechanisms in physical reasoning, *i.e.*, Learning from Intuition (LfI) and Learning from Dynamics (LfD). While it is generally believed that learning the dynamics of physics could help downstream reasoning, a beginning trial of RPIN and ViT on PHYRE-B challenges this fundamental assumption: A ViT model effectively learns to perform physical reasoning without any additional supervision from the ground-truth dynamics, object-centric or not. This counter-intuitive and surprising discovery motivates us to ask whether dynamics play an essential role in physical reasoning. We proceed to answer this question by using Ground-truth Dynamics (GD) from PHYRE's simulator and feeding the sequence to a TimeSformer model. Experimental results show that accurate dynamics can boost problem-solving performance. We further explore why Approximate Dynamics (AD) from dynamics predictors perform unfavorably in physical reasoning. We note that the task-solution model is not the one to blame. In addition, despite increasingly accurate dynamics prediction over the years, we notice that noisy dynamics prediction still has

a negative impact on the overall performance in reasoning; during parallel optimization, the LfD paradigm collapses into LfI. We hypothesize that in the long run, uncertainty in dynamics prediction unavoidably accumulates, leading to the inferior final performance. Finally, we dig deeper into the LfI paradigm and check the performance of various classification models in PHYRE. The experimental results show that these models achieve much better cross-template generalization while remaining competitive in within-template generalization. It turns out LfI can still perform well even if accurate dynamics are hard to predict, providing a route for a simpler, more natural, and less task-specific framework for physical reasoning. However, the LfD route, though challenging, remains a lucrative approach to this problem if dynamics could be predicted in a significantly more accurate manner.

**Why do dynamics-based models struggle to make accurate predictions?**    Analyzing the experimental results, we try to provide preliminary explanations on why current dynamics prediction models struggle. We summarize the following three possible reasons:

1. The dynamics prediction itself is challenging, especially in unseen scenarios. For one thing, prediction into the far future is intrinsically difficult as unforeseeable events might steer the dynamics in another direction. For another, the errors will accumulate from earlier frames, leading to exploding uncertainty and noise in the future. Unfortunately, current dynamics prediction methods cannot produce a robust model to predict accurate dynamics in physical scenes.

2. Pixel-based dynamic representation has more information than object-based representation, while object-based representation is more concise. Arguably, pixel-based representation potentially incorporates all necessary information, such as shapes of objects, potential points of collision, and angular velocity. However, such a representation is extremely noisy, hence extracting useful information embedded is difficult. In comparison, object-based representation is by design concise and follows the general principles in the laws of physics. Nevertheless, object-centric methods lose essential clues in the scenes especially when it comes to collision and its aftereffects. The fact that there is not yet a feature representation method that summarizes all necessary information for physical modeling further complicates physical reasoning.

3. Task-solution model design might play a role, though not significant. The self-attention mechanism has constantly proven rewarding in a variety of tasks. The strong performance of LfI could also benefit from this architectural design.

**Limitations and future work**    Building intelligence with physical reasoning ability is a never-ending journey. However, there are still many aspects left to be studied in the future.

- In contrast to the topic's significance is the lack of rich environments for experiments. Therefore, we only conduct experiments on PHYRE to support our insights. In the future, we hope to develop a suite of physical reasoning tasks and further investigate the insights presented in this work. In addition, we would also like to incorporate the latest advancements in the suite to see if LfI is a more feasible approach to physical reasoning.

- In the experiments, we use original LfI models in our experiments without extra design for physical reasoning tasks. We hypothesize that better perceptual modules and more useful spatial analyzing modules related to physical reasoning could further improve the performance of LfI. In particular, the most challenging physical dynamics in PHYRE involve collision which is a non-smooth and angular movement that does not lie in the traditional linear Euclidean space. We hope to incorporate insights for physical modeling into LfI models for further refinement.

- Analyzing the experiments in Sec. 5, an additional question to ask is how accurate the dynamics prediction needs to be for it to be beneficial. We did not figure out a measure for this problem in this work. Carefully masking or introducing noise to specific image areas and testing physics predictors at different levels may lead to an answer.

- While difficult, the dynamics prediction route remains appealing as indicated by its upper bound and the status quo. We are curious how dynamics will benefit other reasoning processes, such as counterfactual and hypothetical reasoning. We argue the dynamics prediction route is still worth additional efforts and look forward to advances made in this field.

**Societal impacts**    No foreseeable negative societal impacts in this work.

**Acknowledgment**    We would like to thank Miss Chen Zhen at BIGAI for making the nice figures and the anonymous reviewers for their constructive comments.

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
