# OpenReview forum: "On the Learning Mechanisms in Physical Reasoning"
_NeurIPS.cc/2022/Conference — NeurIPS 2022 Accept_

### Official Review · Reviewer_SWzY · 2022-07-07

**Rating:** 7
**Confidence:** 4
**Soundness:** 4 excellent
**Presentation:** 4 excellent
**Contribution:** 3 good

**Summary:**

The paper examines two types of approaches for physical reasoning, namely learning from intuition (LfI) and learning from dynamics (LfD). LfI directly predicts the outcome without trying to learn or predict the trajectory of the process. Surprisingly enough, the paper shows that existing state-of-the-art LfD methods are only as good as LfI. The reason is that LfD methods are very inaccurate for predicting long-term dynamics. These claims are supported by thorough experiments.

**Questions:**

I wonder if the LfD baselines considered are a bit weak. For example, [1] provides a strong method in learning the dynamics of physics scenes. Is LfI really as good as LfD?

[1] Janner, Michael, et al. "Reasoning about physical interactions with object-oriented prediction and planning." arXiv preprint arXiv:1812.10972 (2018).


**Limitations:**

Limitations discussed thoroughly by the author.

**Strengths And Weaknesses:**

$\textbf{Strengths}$

The paper clearly states the research questions and answers them with thorough experiments. Many of the experiments clearly ablate the components and provide convincing conclusions.

The paper is well written with minimal error.

$\textbf{Weaknesses}$

The baselines from LfD seem to be a bit weak and maybe the authors should perform more experiments on stronger LfD baselines.

---

> ### Author Response · Authors · 2022-08-02
> **Response to R4 SWzY**
>
> We are grateful for your positive feedback on our experimental design and paper presentation, as well as for providing us with a potential LfD model for learning helpful dynamics of physics scenes.
> > I wonder if the LfD baselines considered are a bit weak. For example, [1] provides a strong method in learning the dynamics of physics scenes. Is LfI really as good as LfD?
>
> Thanks a lot for the pointer. We have discussed this method in revision (L110). However, we did not consider this baseline for the following reasons:
> - [1]'s physical environment is different from the PHYRE benchmark used in our work.
>   - The environment of block towers has limited object types compared with PHYRE. There are only three types of blocks here, while PHYRE consists of balls, bars, jars, and standing sticks that are even non-regular.
>   - The dynamics types are different. The environment of block towers focuses more on short-term dynamics such as falling and rotation, while PHYRE focuses more on how interactions affect the long-term dynamics.
> - O2P2's dynamics prediction model has a similar structure to RPIN considered in this work
>   - Both of them use an object-based representation and take a pairwise interaction network as the backbone. We consider RPIN in our work due to its proven success in PHYRE while sharing similarities with modern models like O2P2.
>
> Based on our results, current LfI models are indeed just as good as LfD models. Since it is very counterintuitive, we believe our work sheds light on future research directions on physical reasoning and would appeal to a broad audience.

---

### Official Review · Reviewer_8EL5 · 2022-07-11

**Rating:** 6
**Confidence:** 2
**Soundness:** 3 good
**Presentation:** 3 good
**Contribution:** 3 good

**Summary:**

The paper systematically studies the impact of different learning mechanisms on physical reasoning problems through four controlled experiments. The results suggest that (i) reasoning based on a static scene without dynamics predictions can perform as good as that based on an approximate dynamic model subject to errors, (ii) reasoning based on highly accurate dynamics models (Ground-truth
Dynamics) can yield even stronger results.

**Questions:**

It’s noted in Discussion that `object-centric methods lose essential clues in the scenes especially when it comes to collision and its aftereffects`. Why does a dynamic object-centric model lose essential clues?

**Limitations:**

Yes, it’s well discussed.

**Strengths And Weaknesses:**

Strengths
* The paper challenges the common belief on the role of dynamics prediction for physical reasoning through a rich set of empirical experiments. The experiments are very well motivated and designed. Many recent models like ViT, Swin, BEiT, etc. are included in comparison. The results provide strong evidence about the pros and cons of different learning mechanisms on the PHYRE benchmark.
* The paper is very well written. In particular, the detailed discussions on open questions, limitations and future work are inspiring.

Weaknesses
* The final conclusions from this paper seem highly in line with common expectations: good static models might be better than mediocre dynamic models but worse than accurate dynamic models for physical reasoning. I’m not sure if the paper really brings any new insights other than empirical evidence.

---

> ### Author Response · Authors · 2022-08-02
> **Response to R3 8EL5**
>
> Thank you for your positive assessment of our work as well-motivated and inspiring.
> > The final conclusions from this paper seem highly in line with common expectations: good static models might be better than mediocre dynamic models but worse than accurate dynamic models for physical reasoning.
>
> Looking at the experimental results and conclusion, we hope that we have delivered an intuitive and clear case. However, the conclusion here is not as straightforward as a common expectation in the learning community that the review thought. In literature, most of the work assumes by default that the models with dynamics prediction, in almost all cases, are more effective than those without. Thus, researchers almost bet a lot on LfD and ignore the potential of LfI models. In fact, as we have shown in this work, the community of LfD is struggling with the design of accurate dynamics prediction modules and can only create some mediocre dynamic models with the same level of LfI. To reach this conclusion, thorough analyses are required to challenge the previous preference for dynamics prediction modules and attract more researchers to think about the crucial research question on physical reasoning:
>
> **For now, are we in the wrong direction of physical reasoning research by creating deliberately designed but redundant dynamics prediction modules?**
>
> In a nutshell, we believe it is significant to address this point in the current state of research, inspiring the community to reconsider the learning mechanisms in physical reasoning.
> > It’s noted in Discussion that `object-centric methods lose essential clues in the scenes especially when it comes to collision and its aftereffects`. Why does a dynamic object-centric model lose essential clues?
>
> Object-centric methods usually encode the objects based on their properties, such as positions, velocities, and sizes. It may be easy to predict how the objects' states change when there is no collision event (use calibrated Newtonian physics or approximated physics by neural networks). However, detection of collision events can be hard for object-centric methods due to the difficulty of calculating the accurate collision time based only on the vector of properties, especially when the object shapes (balls, bars, jars, and standing sticks) and dynamics (falling, rotation, collision, and friction) are diverse like those in PHYRE. In comparison, pixel-based methods can handle collisions more easily because there is no need to explicitly model the complex collision process, especially when it involves deformation. Only after obtaining accurate detection of collisions can we logically predict the aftereffects.

---

### Official Review · Reviewer_h3js · 2022-07-11

**Rating:** 6
**Confidence:** 4
**Soundness:** 3 good
**Presentation:** 4 excellent
**Contribution:** 2 fair

**Summary:**

The paper tries to understand the utility of learning dynamics in solving physical reasoning tasks. To do this, the authors use the PHYRE benchmark and test a dynamics-based model (trained to predict future observations) and an intuition-based model (trained to predict the answer directly). Through different experiments, the authors find that 1) Intuition-based models outperform dynamics-based models. 2) Dynamics does help when it is accurate 3)  Approximate modelling of dynamics hurts performance 4) Directly predicting the outcomes with different tweaks/heuristics can be very effective.

**Questions:**

- The central question directly seems to stem from a model-free vs model-based approach. How would the authors relate their work to relevant work in this field?
- What is the ideal benchmark that the authors would want to test the relevant questions?

**Limitations:**

The authors do a good job with discussing the limitations of the paper. Others are discussed in the weakness section above.

**Strengths And Weaknesses:**

Strengths

- The paper is clearly written and is easy to understand.
- The motivations, models and questions are well placed in the current literature trying to understand how we could model the physics in the world
- The experiments are well motivated and the successive questions follow well from the previous ones.
- The authors do a good job of testing different models and training regimes throughout the experiments presented in the paper.

Weakness

- The main problem that makes the claims presented in the paper weak is the use of a single benchmark to test hypotheses. This makes it difficult to assess whether the observed effects are an artefact of the benchmark or are more consistent across domains.
- Intuition-based models might be better at the relatively narrow Phyre benchmark (where the goal is to place a ball), and dynamics-based models could be better at answering a broader set of questions. For example, how would the performance of intuition-based models differ when solving tasks that require counterfactual reasoning or hypotheticals? Experiments /discussions on this are missing from the paper.
- Relatedly, dynamics seems to be most useful on out of distribution or systematic generalization. The cross-task split answers this question partially as GD models are able to achieve a cross-task performance similar to within-task. But this split might still not be general enough. For example, one can imagine that a dynamics-based model would find it easier to transfer knowledge when the task is changed slightly, like changing the location of another object in the scene instead of the ball.
- A direct extension of 5.3 is to use Ground truth Dynamics with noise. This would give a more complete answer of how good the dynamics model needs to be to help solve a task.
- Minor: I would have liked to see more of a discussion of relevant cognitive science literature on how humans use simulation of dynamics vs intuition for physical reasoning

---

> ### Author Response · Authors · 2022-08-02
> **Response to R2 h3js**
>
> We feel incredibly thankful for your acknowledging our paper as clear and logical and our experiments as well-motivated. We also thank you for raising those interesting questions.
> > The main problem that makes the claims presented in the paper weak is the use of a single benchmark to test hypotheses.
>
> The lack of such goal-driven physical reasoning benchmarks is a problem facing the entire community. And if the reviewer has good suggestions, we'd be more than happy to consider them. However, we believe PHYRE is representative and currently the only viable benchmark because of its diversity and complexity. It consists of many different object types, complex Newtonian dynamics, and even generalization splits. Thus we believe, for now, PHYRE is the most suitable and the only candidate.
> >Intuition-based models might be better at the relatively narrow Phyre benchmark, and dynamics-based models could be better at answering a broader set of questions. Experiments /discussions on this are missing from the paper.
>
> Good suggestion! To study LfI and LfD, we only consider the most practical goal-driven tasks that directly use physical reasoning to reach a specific state. Counterfactual reasoning and hypotheticals are definitely important and worth mentioning and could well be an interesting topic to consider. We have discussed this point in revision (L354-355 ) and will consider it in future work.
> >Relatedly, dynamics seems to be most useful on out of distribution or systematic generalization. The cross-task split answers this question partially as GD models are able to achieve a cross-task performance similar to within-task. But this split might still not be general enough.
>
> There might be a misunderstanding about the within-task and cross-task generalization settings. In the within-task setting, the test sets are the tasks that changed slightly, like the locations of the background objects. In a cross-task setting, the test sets are the tasks that have a completely different layout of objects, not just the slight location change. The cross-task is considered more difficult than within-task. The generalization splits should be general and challenging enough. But let us know if we misinterpret your question.
> > A direct extension of 5.3 is to use Ground truth Dynamics with noise. This would give a more complete answer of how good the dynamics model needs to be to help solve a task.
>
> Thanks for the suggestion. In this work, we present a thorough analysis of where we are now from the boost of dynamics prediction and would love to consider this aspect in future studies (L349-352).
> > I would have liked to see more of a discussion of relevant cognitive science literature on how humans use simulation of dynamics vs intuition for physical reasoning.
>
> Thanks for your valuable advice. In the cognitive study of how humans predict the drop point of a ball released from a pendulum [1], both the experimental design and the discovery share great similarities with our work. Participants are tested in two ways: 1) predicting the drop point directly or 2) predicting the drop point after drawing the ballistic trajectory. The former corresponds to the method of LfI while the latter corresponds to LfD. The results show that humans make a more accurate estimation by predicting directly, which also corresponds with our conclusion that LfI reaches or even outperforms LfD.
> [1] Different physical intuitions exist between tasks, not domains, Computational Brain & Behavior 2018
> > The central question directly seems to stem from a model-free vs model-based approach. How would the authors relate their work to relevant work in this field?
>
> Model-free and model-based are the most common terms in reinforcement learning; it is a reasonable analogy. LfI, is a direct learning manner without prior knowledge about how the states will change in the physical environment, which shares similarities with the model-free methods. LfD first constructs the physical dynamics model, similar to the model-based methods that assume the environment can be learned. Humans are somewhere in between; we do short-term and inaccurate simulations but still manage to solve the problems.

---

> > ### Author Response · Authors · 2022-08-02
> > **Response to R2 h3js (part 2)**
> >
> > > What is the ideal benchmark that the authors would want to test the relevant questions?
> >
> > The physical reasoning task PHYRE provides complicated interactions of multiple objects and abundant scenes to guarantee its ability to test a model's performance on difficult tasks and the generalization of novel scenarios. However, aimed at diving deeper into the research of two learning mechanisms, the ideal benchmark can include:
> > - Different time lengths of a task: whereas playing one PHYRE task usually takes 10-15 seconds, an ideal benchmark can contain different tasks of multiple time lengths, which will lead to further discussion of how LfD performs under different numbers of the predicted frames.
> > - While PHYRE is based on 2D dynamics, the ideal benchmark can expand to 3D or even be based on the real-world scene while reserving the complexity and variety of PHYRE.
> > - The reward of PHYRE is binary, and there is only one winning condition per task, while an ideal benchmark can return a continuous number from 0 to 1, or return different rewards according to different conditions, which may help to figure out the internal mechanism of LfI.
> > - We also expect novel evaluation metrics; AUCCESS might encourage extensive sampling and evaluation, while the ideal one is measured on how many actions are needed until the problem is solved. Online learning might get involved in this process.

---

> > > ### Comment · Reviewer_h3js · 2022-08-09
> > > **Response to authors**
> > >
> > > Thank you for your detailed response in answering my questions and concerns. After reading the other reviews and author responses, I will keep my current score!

---

### Official Review · Reviewer_TKqA · 2022-07-11

**Rating:** 7
**Confidence:** 4
**Soundness:** 3 good
**Presentation:** 3 good
**Contribution:** 3 good

**Summary:**

The paper compares Learning from Dynamics (LfD) and Learning from Intuition (LfI) in the context of physical reasoning. Through four sets of experiments, it concludes that the state-of-the-art LfI method is at least as good as LfD while being much simpler. It also sheds light on potential improvements in LfD methods.

**Questions:**

Does the paper provide a position on the preference between LfI and LfD? The experimental results seem to suggest LfI is simpler and generalizes better. But it's also mentioned in the paper that "the potential improvement from LfD, though challenging, remains lucrative".



**Limitations:**

Lack of rich experimental environments in the experiments (upfront). There're existing environments that would provide a more comprehensive study, such as [1-4]. This could also explain why better image classification models used in Sec 5.4 did not yield better AUCCESS.





[1] A. Dosovitskiy, G. Ros, F. Codevilla, A. Lopez, and V. Koltun. Carla: An open urban driving simulator. In Proceedings of the Annual Conference on Robot Learning, pages 1–16, 2017.
[2] M. Savva, A. Kadian, O. Maksymets, Y. Zhao, E. Wijmans, B. Jain, J. Straub, J. Liu, V. Koltun,J. Malik, D. Parikh, and D. Batra. Habitat: A platform for embodied ai research. arXiv preprint arXiv:1904.01201, 2019.
[3] E. Todorov, T. Erez, and Y. Tassa. Mujoco: A physics engine for model-based control. In Proceedings of the International Conference on Intelligent Robots and Systems, 2012.
[4] Y. Wu, Y. Wu, G. Gkioxari, and Y. Tian. Building generalizable agents with a realistic and rich
3D environment. In arXiv:1801.02209, 2018.

**Strengths And Weaknesses:**

### Strength
A thorough systematic comparison between LfI and LfD: It challenges the common assumptions in improving dynamics prediction and designed four experiments subsequently. It provided detailed analysis for each experiment, with insightful observations such as challenges in accurate dynamics prediction.

The code is provided and is also well documented.

### Weakness
Experimental results were not repeated (such as to calculate the mean and variance of the accuracy rate) thus its statistical significance can be challenged.

---

> ### Author Response · Authors · 2022-08-02
> **Response to R1 TKqA**
>
> First of all, thanks for recognizing our work to be thorough and insightful! See below for answers to your questions.
> >Experimental results were not repeated, thus its statistical significance can be challenged.
>
> Thanks for the reminder. Due to time constraints and the sheer volume of all experiments, we repeated experiments on the part that most supports our argument (the trials of LfI models in Section 5.4): LfI is a much simpler yet more effective way compared with LfD. Specifically, we train each of our three LfI models three times by varying the random seed. See Table 4 in the revised manuscript.
> > Does the paper provide a position on the preference between LfI and LfD?
>
> Based on empirical studies, we note that LfI is a much simpler yet more effective way to conduct physical reasoning. But LfD could be potentially helpful were the following issues addressed:
> - Accurate long-term dynamics prediction
> - Novel task-solution model design that is more tolerable to imprecise dynamics
>
> Our goal is to investigate recent works that simply add prediction modules without further studying the reasoning process, the two mechanisms (LfI and LfD), as well as future directions.
> > Lack of rich experimental environments in the experiments. There're existing environments that would provide a more comprehensive study, such as [1-4].
>
> Thanks for pointing them out, and these works are definitely worth discussing in our revision (L102-105)! However, we also note that there are important discrepancies between them and the physical environments which can verify our argument in this paper:
> 1. Carla is mainly for domains of self-driving and involves social norms and complex visual understanding, which are beyond the scope of physical reasoning in our work.
> 2. Habitat and House 3D are introduced for embodied agents, which require the combination of vision, language, and robotics, instead of focusing on physical reasoning, and if any, physics in them is too simplified. These tasks are primarily used for navigation.
> 3. MuJoCo is more for control rather than conducting long-term physical reasoning and applying physical laws to solve problems.

---

### Meta-Review · Area_Chair_DdUV · 2022-08-29

**Recommendation:** Accept
**Confidence:** Certain

**Metareview:**

This paper investigates a simple and important question: does learning to predict physical dynamics help an agent perform better physical reasoning? While most prior work automatically treats this as a given, the paper provides interesting findings that intuition-based learning is better than dynamics-based learning, especially when the dynamics model is approximate. All the reviewers appreciated the clear writing and thorough experiments performed. I believe this will be an insightful and impactful paper.

**Award:**

No

---

### Decision · Program_Chairs · 2022-09-14

Accept